# Advances of H_2_S in Regulating Neurodegenerative Diseases by Preserving Mitochondria Function

**DOI:** 10.3390/antiox12030652

**Published:** 2023-03-06

**Authors:** Lina Zhou, Qiang Wang

**Affiliations:** Departments of Anesthesiology, Center for Brain Science, The First Affiliated Hospital of Xi’an Jiaotong University, Xi’an 710061, China

**Keywords:** hydrogen sulfide, oxidative stress, mitochondria dysfunction, sulfhydration, neurodegenerative diseases

## Abstract

Neurotoxicity is induced by different toxic substances, including environmental chemicals, drugs, and pathogenic toxins, resulting in oxidative damage and neurodegeneration in mammals. The nervous system is extremely vulnerable to oxidative stress because of its high oxygen demand. Mitochondria are the main source of ATP production in the brain neuron, and oxidative stress-caused mitochondrial dysfunction is implicated in neurodegenerative diseases. H_2_S was initially identified as a toxic gas; however, more recently, it has been recognized as a neuromodulator as well as a neuroprotectant. Specifically, it modulates mitochondrial activity, and H_2_S oxidation in mitochondria produces various reactive sulfur species, thus modifying proteins through sulfhydration. This review focused on highlighting the neuron modulation role of H_2_S in regulating neurodegenerative diseases through anti-oxidative, anti-inflammatory, anti-apoptotic and S-sulfhydration, and emphasized the importance of H_2_S as a therapeutic molecule for neurological diseases.

## 1. Introduction

Neurodegenerative diseases in both humans and animals are usually complicated. They can be induced by pharmaceuticals and various environmental chemicals, including antiviral drug tilorone, typhaneoside, metronidazole, homocysteine, paraquat and some heavy metals [1]. In the brain, mitochondria with striking structure are mainly responsible for cellular energy production, which occurs through oxidative phosphorylation [2]. They are the primary target for toxin-induced bioenergetic failure, subsequently resulting in oxidative stress and mitochondrial dysfunction [3]. Accordingly, oxidative damage has long been recognized as the initial cause of neurodegenerative diseases, including Alzheimer’s disease (AD), Huntington’s disease (HD) and Parkinson’s disease (PD) [4,5]. In addition, mitochondrial dysfunction has been described as a pathological hallmark of neurotoxicity development and cognitive impairment [5,6]. Thus far, the mechanisms involved in modulating cellular redox homeostasis and protecting mitochondrial function during neurotoxicity, and efficient mechanism-based therapies have not been fully elucidated.

In recent years, hydrogen sulfide (H_2_S) has been accepted as a novel gas transmitter, followed by nitric oxide (NO) and carbon monoxide (CO). The earliest discovery of H_2_S in 1989 reported the endogenous production of H_2_S and its bioactive properties in mitochondria fraction of rats’ brain [7], implying that H_2_S may exert physiological functions in the mammalian central nervous system (CNS). Numerous studies have reported that H_2_S can be endogenously produced by three main enzymes in mammalian cells, i.e., cystathionine β synthase (CBS), cystathionine γ lyase (CSE), and 3-mercaptopyruvate sulfur transferase (3-MST) [8]. Moreover, a newly reported enzyme, D-amino acid oxidase (DAO), has been confirmed to promote H_2_S production in mitochondria [9]. The molecular mechanism mediated by H_2_S is the S-sulfhydation, which modifies the thiol groups of specific cysteine residues of target proteins, resulting in alterations of protein structure, enzymatic activity, or translocation [10]. The reactive sulfur species required for posttranslational modification mediated by protein persulfidation are generated from mitochondria through H_2_S oxidation [11]. The cytoprotective functions of H_2_S reported in mammals mainly include antioxidant defense, autophagy induction, neurotransmission, vasorelaxation, lifespan extension, etc. [12,13,14,15]. In 1996, Kimura et al. reported that H_2_S functions as a neuromodulator to induce LTP (long-term potentiation) via activating NMDA receptors [16]. Furthermore, numerous studies on the pharmacological and physiological role of H_2_S characterized H_2_S as having anti-inflammatory and antioxidant properties in brain neuron systems [17,18]. It has been recently reported that H_2_S protects animals from homocysteine-induced neurodegeneration by attenuating oxidative DNA damage, mitochondrial disorders and mitochondria-mediated apoptosis [19]. Collectively, research of H_2_S-mitochondrial in the brain advanced our understanding of mechanisms of neuron diseases, and specific compounds releasing H_2_S into mitochondria could be developed as therapeutic molecules for different neurodegenerative diseases in the future.

This review mainly focused on H_2_S as a protective gaseous signaling molecule in the development of neurodegenerative diseases. First, we provided an overview of neurodegenerative diseases associated with mitochondria dysfunction and oxidative stress. Furthermore, we highlighted the biological effect and regulatory mechanisms of H_2_S on different neurodegenerative diseases. Finally, we provide perspectives on potential therapeutic treatment using H_2_S-releasing drugs in modulating neurological diseases.

## 2. Induction of Neurotoxicity

Different toxic substances, including natural products, environmental chemical compounds, and pharmaceuticals, can affect the nervous system, inducing neurotoxicity in humans and laboratory animals. Micronutrient homocysteine (Hcy) is a sulfur-containing amino acid derived from methionine metabolism. The earliest study of Hcy-induced neurotoxicity was reported in patients with a deficiency of cystathionine beta-synthase (CBS) [20,21]. It was later found that the CBS enzyme was mainly expressed in the human brain [22]. It was also found to have the ability to convert Hcy into cysteine and glutathione; thus, Hcy is accumulated in the brain under a deficiency of CBS. This was further confirmed in *CBS* knockout mice (Cbs−/+ or Cbs−/−), in which the Hcy level increased approximately 2–50-fold compared with wild-type mice, leading to oxidative stress and neuronal death [23,24]. An accumulating body of literature has reported on neurotoxic mechanisms induced by Hcy. In rat ischemic brain cells, accumulation of Hcy influences mitochondrial ultrastructure, mitochondrial complex I-III enzymatic activities and phosphorylation of mitochondrial STAT3 (mitoStat3), resulting in mitochondrial injury and oxidative stress, ultimately leading to neurotoxicity [25]. Similarly, impairment of mitochondrial activity and oxidative damage were also considered as the prime mechanisms of neurodegeneration in the Hcy-induced Parkinson’s disease (PD) rat model [5]. These findings further indicated that mitochondrial dysfunction is the major cause of different neurodegenerative diseases. In addition, excess Hcy can accumulate cytokine levels in the brain and disturb the inflammatory system [26]. Furthermore, hyperhomocysteinemia (HHcy) induces excitotoxicity and cognitive impairment by disturbing redox potentials and activating N-methyl-D-aspartate (NMDA) receptor [27]. Meanwhile, HHcy was found to impair activity of cytochrome c oxidase (COX) in mitochondria and induce reactive oxygen species (ROS) accumulation, subsequently resulting in apoptotic cell death and neurological dysfunction in primary neurons of rats and humans [28]. Thereby, oxidative damage and mitochondrial dysfunction are considered the most important pathomechanisms in Hcy/HHcy-induced apoptosis and neurotoxicity.

Glutamate is a major excitatory neurotransmitter in the brain. Glutamate-induced excitotoxicity mainly includes calcium imbalance and mitochondrial swelling [29]. It was reported that typhaneoside, a flavonoid compound, suppresses glutamate release in rat cerebrocortical nerve terminals by inhibiting voltage-dependent calcium entry [30]. Glutamate accumulation can over stimulate NMDA receptors, leading to excitotoxicity and disruption of learning and memory [31]. Meanwhile, treatment of the herbicide paraquat promotes the accumulation of β-amyloid and tau protein [32], which have been recognized as pathophysiological events associated with the development of AD.

Organophosphorus pesticides (OPs), which are widely used for controlling pests worldwide, induce neurotoxicity by disturbing mitochondrial function and inducing oxidative stress [33]. Arsenic is a toxic metalloid widely distributed in groundwater. It has been found that arsenic exposure induces neurotoxicity and cognitive impairment by damaging mitochondrial biogenesis [34,35]. Aluminium is a toxic metal ubiquitously present on earth that impairs mitochondrial bioenergetics and leads to cognitive disorder once it enters the brain [36]. Moreover, it has been found that N-methyl-4-phenylpyridinium (MPP^+^) related compounds are dopaminergic toxins that can induce mitochondrial oxidative stress and dysfunction [37]. Thus, they are usually used to induce animal models of PD. Generally, these compounds disturb intracellular redox status and mitochondrial activity, ultimately leading to neuronal death and cognitive impairment. Hence, preserving cellular redox homeostasis and mitochondria function can be regarded as the major therapeutic target for different neurodegenerative diseases.

## 3. Biosynthesis and Metabolism of H_2_S

### 3.1. Physiological Function of H_2_S

While H_2_S is colorless, it is characterized by an offensive smell of rotten eggs [38]. It can freely penetrate cell membranes due to the lipid solubility property [39]. At avery early stage, H_2_S was considered toxic due to the inhibition of mitochondria cytochrome c oxidase, which causes respiratory depression, pulmonary edema and central neuronal system damage [7,40,41,42]. However, in 1989 and 1990, high levels of sulfide in the brain of rats, humans, and bovines were detected, indicating they possibly have a neuromodulator role of H_2_S in neurons [7,42,43]. In 1996, it was found that the appropriate application of H_2_S donor sodium hydrosulfide (NaHS) facilitates hippocampal LTP by enhancing the activity of the NMDA receptor [16]. These findings preliminarily suggested that H_2_S acting as a novel signaling molecule may have a physiological role in modulating neuronal activity.

In mammalian cells, pyridoxal-5′-phosphate (PLP)-dependent enzymes cystathionine-β-synthase (CBS) and cystathionine-γ-lyase (γ-cystathionase CSE) have important roles in generating H_2_S from L-cysteine (Cys) (Figure 1) [16,38]. Recently, it was reported that 3-mercaptopyruvate sulfurtransferase (3-MST) could also produce H_2_S from L-cysteine in combination with cysteine aminotransferase (CAT) in brain nervous system (Figure 1) [44]. Moreover, a new pathway involved in H_2_S generation in mammals revealed that peroxisome-dependent DAO uses D-cysteine to produce 3MP, and subsequently producing H_2_S in mitochondria (Figure 1) [9]. Except for enzyme-dependent production of H_2_S, it was also found in 2009 that H_2_S can be stored as bound sulfane sulfur by adding sulfur to specific cysteine residues of target proteins, which may have an important role in the response to physiological stimuli via releasing H_2_S in the brain [45,46,47]. Thereby, H_2_S and polysulfides may function as signal molecules in the nervous system.

The above-mentioned enzymes responsible for H_2_S production are tissue-specific enzymes involving certain biochemical pathways in humans [48]. The expression of CBS, which is mainly responsible for H_2_S production in the brain, is higher in the hippocampus and cerebellum than the CSE. It is especially expressed in radial glia and astrocyte in the central neuron system of the brain [22,49]. This is further confirmed by the evidence that H_2_S production remains unchanged in the brain through using CSE inhibitors D, L-propargylglycine (PAG) and β-cyano-L-alanine (β-CNA), although they suppress H_2_S generation in the liver and kidney [16,50]. Moreover, it has been found that CBS knockout mice had abnormal development of cerebellar morphology [49]. Additionally, the 3-MST is also mainly localized in brain tissues, including hippocampal pyramidal neurons, mitral cells in the olfactory bulb and cerebellar Purkinje cells [44]. Because the H_2_S produced by 3-MST/CAT is mainly localized to mitochondria, it can attenuate mitochondria dysfunction and protect cells from oxidative stress by scavenging reactive oxygen species [6]. Moreover, D-cysteine-dependent H_2_S production preferentially operates in the cerebellum and the kidney via attenuating oxidative stress [9]. Thus, these discoveries suggest that H_2_S acts as an antioxidant by regulating cellular redox status in the brain.

It is well documented that H_2_S is toxic at high concentrations and beneficial at low concentrations. These investigations emphasize the importance of identifying a strict fine balance of physiological H_2_S for therapeutic use. The endogenous steady-state level of H_2_S is mainly dependent on its oxidation and metabolism in mitochondria. In the mitochondrial matrix, sulfane sulfur is generated from sulfide quinone oxidoreductase (SQR)-mediated H_2_S oxidation [51,52,53]. This reaction forms persulfide (R-SSH) and releases two electrons to the electron transport chain (ETC) via coenzyme Q (CoQ). After that, ethylmalonic encephalopathy 1 (ETHE1) oxidizes the persulfide to generate sulfite, which is further oxidized to SO_4_^2−^ by SO or to S_2_O_3_^2−^ by TST (Figure 1) [54]. In addition to the oxidation of H_2_S, H_2_S can also be catabolized through methylation via thiol S-methyltransferase (TSMT) and the H_2_S-scavenging pathway via metalloproteins [55,56]. More importantly, the H_2_S-mediated transsulfuration pathway can also integrate with other sulfide-linked metabolic pathways, including the folate cycle, glutathione system, and nucleotide metabolism [57]. Thus, the cellular H_2_S levels are tightly controlled via H_2_S biosynthesis and catabolism pathway in order to maintain its physiological protective effect and cellular metabolic profile.

H_2_S synthesis are mainly catalyzed by the enzymes CBS, CSE and 3-MST. DAO is a newly discovered enzyme that can produce H_2_S in mitochondria. H_2_S oxidation occurs in mitochondria, and H_2_S is oxidized to persulfide (SQR-SSH) by SQR via accepting a thiol (-SH), after which the persulfide is oxidized to SO_3_^2−^ by ETHE1. The SO_3_^2−^ is oxidized to SO_4_^2−^ and S_2_O_3_^2−^ by SO and TST, respectively. In addition, H_2_S is methylated by thiol S-methyltransferase (TMST) to form methanethiol and dimethyl sulfide. The dimethyl sulfide is oxidized to SO_4_^2−^ by rhodanese. Transsulfuration pathway (TSP), which involves the transfer of sulfur from homocysteine to cysteine, can integrate with sulfide-associated metabolic pathways, including methionine cycle, folate cycle, glutathione synthesis and nucleotides metabolism to maintain a physiological balance of H_2_S. H_2_S, hydrogen sulfide; CBS, cystathionine β-synthase; CSE, cystathionine γ-lyase; CAT, cysteine aminotransferase; DAO, D-amino acid oxidase; 3MST, 3-mercaptopyruvate sulfurtransferase; SQR, sulfide quinone oxidoreductase; ETHE1, ethylmalonic encephalopathy 1; TST, thiosulfate sulfurtransferase; SO_3_^2−^, sulfite; SO_4_^2−^, sulfate; S_2_O_3_^2−^, thiosulfate; TSP, transsulfuration pathway. The pink arrows indicate the transsulfuration pathway (TSP) and related metabolic pathways. This model was created using BioRender (https://biorender.com/, accessed on 12 February 2023).

### 3.2. Donors of H_2_S

H_2_S donors NaHS and disodium sulfide (Na_2_S) can quickly release H_2_S and display cytoprotective effect by regulating mitochondrial biogenesis and function [58,59]. GYY4137 is believed to release H_2_S dependent on acidic pH and higher temperatures [60]. It can protect mitochondria and vascular endothelial cells from oxidative damage [61]. Furthermore, S-adenosylmethionine (SAM) is regarded as an activator of endogenous H_2_S generation, which could inhibit endothelial growth factor-A-related diseases [62]. Besides biochemical synthesized H_2_S donors, H_2_S can also be generated from natural plants. For example, diallyl disulfide (DADS) and diallyl trisulfide (DATS) derived from Allium plant garlic function as H_2_S donors, impairing the mitochondrial function [63] or decreasing the ROS production in mitochondria [64]. Concerning the contributions of H_2_S donors to clinical experiments, SG1002, a novel H_2_S prodrug, was clinically used in patients with heart failure. It was found that H_2_S is essential for vascular homeostasis by preserving mitochondrial functions and increasing myocardial vascular density [65,66]. MZe786, the ADTOH H_2_S donor, could improve preeclampsia state through lowering blood pressure and attenuating renal damage in mice model C57Bl/6 J of preeclampsia [67,68]. Meanwhile, the administration of H_2_S can protect mitochondrial activity against a high level of antiangiogenic-factor-sFlt-1-induced mitochondrial respiration inhibition and superoxide production in endothelial cells [69]. H_2_S producer sodium thiosulfate exhibits anti-inflammatory and antioxidative effects and has already been investigated in phase I trials for potential benefits in patients with an acute coronary syndrome undergoing coronary angiography [70].

Recently, it has been reported that both AP39 and AP123 are mitochondrial-targeted H_2_S donors whose application can regulate glucose-oxidase-induced oxidative stress and improve mitochondrial function in endothelial cells [71]. It has also been reported that both AP39 and AP123 can be targeted for mitochondrial decrease hyperpolarization of the mitochondrial membrane and increase the electron transport at respiratory complex III to improve cellular metabolism [72]. Thereby, mitochondria-targeted H_2_S donors can potentially be used in the treatment of neurological diseases in the near future.

## 4. Involvement of H_2_S in Neurological Diseases

The physiological levels of H_2_S in human brain tissue are nearly 50–160 μM, which suggests that H_2_S may exert a possible biological function in neurons [16]. Subsequently, numerous researchers have reported that the possible physiological functions of H_2_S in the brain mainly include potentiating long-term potential (LTP) through activation of the NMDA receptors [16], thus inhibiting mitochondrial dysfunction and oxidative damage by directly or indirectly scavenging free radicals and reactive species [73,74,75,76,77], activating Ca^2+^ influx waves in astrocytes [78], initiating anti-neuroinflammatory responses in both astrocyte and microglia [79,80,81], and modulating apoptotic response signaling in neurons [81,82]. These studies highlight the potential neuroprotective role of H_2_S in regulating neurodegenerative diseases through anti-oxidative, anti-inflammatory, or anti-apoptotic effects (Table 1).

### 4.1. Alzheimer’s Disease (AD)

AD has been characterized by memory deterioration and cognition impairment, mainly caused by neuropathologic hallmark events, which include enlargement of β-amyloid plaque size and hyperphosphorylation of tau protein in the brain [83]. AD-related neurodegeneration is associated with oxidative stress [77]. It has been implied that soluble oligomers of Aβ can deregulate H_2_S signaling by inhibiting cysteine transporter excitatory amino acid transporter 3 (EAAT3/EAAC1), resulting in oxidative stress and neurodegeneration [84,85]. Thereby, H_2_S may have a vital role in AD. Indeed, it was found that the H_2_S levels in both plasma and the brain of AD patients were much lower than those in healthy individuals [86,87]. Furthermore, the plasma H_2_S level was negatively correlated with the severity of AD [87], because H_2_S is synthesized from homocysteine via CBS catalyzation in the transsulfuration pathway, and SAM is the activator of CBS [62]. In AD patients, the homocysteine level in the brain is high, and SAM content is low, whereas the expression level of CBS showed no difference between AD patients and the normal group [86]. Therefore, it has been concluded that the reduction in H_2_S might be associated with decreased activity of CBS [86,88], which further confirms the essential neuroprotective role of the CBS-H_2_S signaling axis in the AD brain. Thus, dietary adjustment has been proposed to improve symptoms of AD patients, as a diet rich in taurine, cysteine, folate, B12, and betaine could promote H_2_S synthesis in the brain [89].

Mitochondrial dysfunction is characterized as a pathogenesis of AD. Previous studies found that H_2_S protected PC12 cells against Hcy or formaldehyde (FA)-induced neurotoxicity by preserving mitochondrial membrane potential (MMP) and attenuating intracellular ROS accumulation [90,91]. AP39, a mitochondrially targeted H_2_S donor, has been found to exert a neuroprotective effect in a dose-dependent manner in APP/PS1 neurons by increasing energy production and cell viability, protecting mitochondrial DNA, and decreasing ROS accumulation at 100 nM concentrations, which reduces Aβ deposition in the brain, inhibits brain atrophy, and ameliorates memory deficiency in the Morris water maze experiment [92]. Meanwhile, the in vitro and in vivo experiments in mammals demonstrated that H_2_S-mediated sulfhydration of ATP synthase helps to maintain mitochondrial bioenergetics [93], implying the possible mechanism of H_2_S in modulating AD development. Thus, it is possible that the exogenous application of appropriate concentration of H_2_S preserves mitochondrial dysfunction in AD by improving mitochondrial redox homeostasis and maintaining energetic production.

Meanwhile, it was found that H_2_S treatment alleviated the excitotoxicity-triggered oxidative stress by decreasing levels of malondialdehyde in the cerebral cortex of the 3xTg-AD mice model [94,95]. Moreover, a novel neuroprotective mechanism of H_2_S has been explored, revealing that H_2_S exerts antioxidative effects by activating nuclear factor erythroid-2-related factor 2 (Nrf2), heme oxygenase-1 (HO-1) and glutathione S-transferase (GST) in APP/PS1 transgenic mice [96]. In addition, Hyperhomocysteinemia (HHcy) has been recognized as a potential risk factor for AD development. It was found that NaHS and MK801 treatments attenuate the high level of the homocysteine-induced blood–brain barrier (BBB) disruption, synaptic dysfunction and excitotoxicity in mice brains by modulating NMDA receptors [97]. It has also been reported that ACS6, an H_2_S donator, preserves mitochondrial function and prevents Hcy-induced apoptosis by inhibiting cytochrome C releasing, ROS accumulation, and caspase-3 activation in PC12 cells of the AD mouse model [98].

H_2_S exerts its anti-inflammatory effect via several signal pathways, including preserving mitochondrial function in a p38 and JNK-MAPK dependent manner in microglia [99], inhibiting activation of NF-κB pathway in the hippocampus [100], suppressing phosphorylation of signal transducer and activator of transcription 3 (STAT3) and cathepsin S (Cat S) activation [101] in mouse models of AD. In addition, H_2_S exerts protective effects through different neuroprotective pathways. For example, it was found that the application of NaHS and Tabiano’s spa water (rich in H_2_S) can slow down the progression of learning and memory impairment in three experimental models of AD, including brain injection of β-amyloid1-40 (Aβ) or streptozotocin-induced rat models, and in an AD mouse model harboring human transgenes APPSwe, PS1M146V and tauP301L (3xTg-AD mice), by lowering phosphorylation level of tau protein, preventing oxidative and nitrosative stresses in the cerebral cortex, upregulating Bcl-2 and downregulating BAX in the hippocampus [95]. Moreover, it has been confirmed that administration of an H_2_S donor (NaHS) into APP/PS1 transgenic mouse improves spatial memory acquisition through shifting from the plaque-forming β pathway to the non-plaque forming α pathway of APP cleavage, a non-amyloidogenic processing of APP [102]. It was also reported that H_2_S decreased Aβ deposition in mitochondria by inhibiting γ-secretase activity [103]. Overall, chronic treatment with H_2_S donor could clearly reduce the size of β-amyloid plaques, and reduce activities of c-jun N-terminal kinases (JNK), extracellular signal-regulated kinases and p38 involved in tau phosphorylation, inflammatory response and apoptosis in the cortex and hippocampus, thus finally protecting AD mice against cognitive impairment [104]. These discoveries show promising therapeutic prospects for H_2_S by targeting multiple pathophysiological events in AD.

Although H_2_S has been shown to exert a neuroprotective effect on account of its antioxidative, anti-inflammatory, and anti-apoptotic properties, and that administration of H_2_S donors is beneficial for AD, the exact molecular mechanisms underlying the neuroprotective effect are still largely unknown. Glycogen synthase kinase 3β (GSK3β) promotes hyperphosphorylation of tau protein, which finally leads to AD (Figure 2). Recently, it has been reported that H_2_S-mediated sulfhydration modifies GSK3β, eventually inhibiting tau’s hyperphosphorylation, which is a major contributor to AD development (Figure 2) [105]. Thereby, the application of H_2_S donor GYY4137 improves cognitive deficiency in the 3xTg-AD mouse model [105]. In summary, in comparison to a normal mouse, the H_2_S level was decreased, and sulfhydration was also diminished in the brain of the AD mouse model [88,105], implying the important role of H_2_S-mediated post-translational modification in regulating AD development.

GSK3β binds to Tau protein and hyper-phosphorylates Tau, which subsequently leads to AD pathology. The application of H_2_S sulfhydrates the GSK3β and inhibits the hyper-phosphorylation of Tau, which finally confers neuroprotection. The red fonts represent sulfhydryl group. This model was created using BioRender (https://biorender.com/, accessed on 19 November 2022).

### 4.2. Parkinson’s Disease (PD)

PD has been characterized by the loss of dopaminergic neurons in the substantia nigra (SN) and dopamine deficiency in the striatum [106]. Oxidative stress, mitochondrial dysfunction, misfolded protein accumulation, and neuroinflammation are considered major pathogenesis contributors to PD, which eventually result in impaired movement [106,107]. Last few decades, a large number of discoveries have indicated that H_2_S has an essential anti-oxidative effect in dealing with PD patients suffering from oxidative stress [76,108,109]. These findings demonstrate that H_2_S exerts its neuroprotective effects by acting as an antioxidant in the brain of PD patients.

In addition, rotenone is used to establish models of PD disease in animals. It has been reported that H_2_S can attenuate rotenone-induced cell apoptosis by preserving mitochondrial function and regulating the JNK-MAPK pathway [82], which suggests that appropriate administration of NaHS is beneficial for PD treatment, and it can be regarded as a potential therapeutic strategy for neurodegenerative disease PD. Indeed, in 6-hydroxydopamine (6-OHDA)-induced PD rat model revealed that endogenous H_2_S level was decreased in the SN. On the other hand, the exogenous application of H_2_S promoted the increase in endogenous H_2_S and inhibited the movement disorder by reversing the depletion of tyrosine–hydroxylase-positive neurons in the SN. This protective effect may be mediated by promoting leptin signaling and increasing malondialdehyde levels in the striatum [110,111]. NaHS application could also protect SH-SY5Y cells against 1-methyl-4-phenylpyridinium ion (MPP^+^)-induced mitochondrial transmembrane potential loss, oxidative stress and cell apoptotic in the PD cell model [109]. Accordingly, inhibition of H_2_S production from the CBS-H_2_S axis enhances 1-methy-4-phenyl-1,2,3,6-tetrahydropyridine (MPTP) or MPP^+^-induced neurotoxicity in the PD model [112,113]. Moreover, injection of GYY4137 (an H_2_S donor) or overexpression of CBS in the striatum attenuates hallmark pathological event of nitrated α-synuclein in the PD model and rescues motor disorder induced by MPTP [112,114]. Since the N-terminus of α-synuclein serves as a mitochondrial targeting sequence peptide and is associated with the regulation of mitochondrial function, it is possible that H_2_S application may preserve mitochondrial activity and cell viability in the brain of PD by attenuating nitrate stress induced by post-translational modification of α-synuclein.

Proper endogenous H_2_S production by strictly controlling CBS synthesis in astrocytes exhibits anti-inflammatory and neuroprotective effects [80]. It was reported that the application of H_2_S inhibited microglial activation in SN and prevented the accumulation of inflammatory factors in the striatum in PD model induced by rotenone [110]. However, recent studies have pointed out that excessive H_2_S generated from gut bacteria promotes the formation of α-synuclein fibrils by releasing cytochrome C from mitochondria and stimulating ROS accumulation [115], thus suggesting that H_2_S may contribute to PD disease. According to these findings, H_2_S exerts an essential physiological role in the brain, mainly depending on strictly controlling physiological H_2_S concentrations. Except for the application of H_2_S-releasing donors, it has been reported that inhaled H_2_S protects MPTP-induced movement disorder in PD mouse model against neurodegeneration via antioxidative, anti-inflammatory, and antiapoptotic signaling in the brain [116], which was further confirmed by the possible beneficial link between chronic inhalation of H_2_S in animal models and protection of dopaminergic neurons associated with PD [117].

The exact mechanism of H_2_S has been defined as the sulfhydration of specific target proteins. It has been recently reported that H_2_S increases expression level of silent information regulator-1 (sirtuin 1, SIRT1), and enhances the activity of SIRT1 through H_2_S-mediated sulfhydration, which could increase autophagy flux and attenuate cell injury in PD mouse model induced by MPP^+^ [118]. Although the SIRT1 is primarily expressed in the cell nucleus, its activity is largely associated with mitochondrial biogenesis and turnover by mitophagy [119]. Peroxisome proliferator-activated receptor γ co-activator 1α (PGC-1α), a metabolic coactivator, can be deacetylated by SIRT1 to modulate mitochondrial biogenesis and respiration [120]. Thus, it is possible that H_2_S-induced sulfhydration of SIRT1 promotes the deacetylation of PGC-1α, finally resulting in the amelioration of mitochondrial activity and improvement of PD (Figure 3A). Furthermore, it has been reported that the parkin protein, a neuroprotective ubiquitin E3 ligase, is depleted in the brains of patients affected by PD. Interestingly, it has been found that the parkin protein can be sulfhydrated by H_2_S, subsequently enhancing its activity (Figure 3B) [121]. Thus, modulating specific protein activities via H_2_S-mediated sulfhydration may be therapeutic in PD disease.

### 4.3. Huntington’s Disease

HD is characterized by damage to the corpus striatum of the brain, leading to involuntary movements and motor disorders along with psychiatric disturbances [122]. It occurs due to the gene encoding huntingtin (Htt) mutation, which results in the expansion of polyglutamine repeats and subsequent cellular processes, including oxidative stress, mitochondrial dysfunction, neurotoxicity, and behavioral dysfunction [123,124]. Previous studies reported that cystathionine γ-lyase (CSE) depletion could contribute to HD pathophysiology [124,125], considering that CSE promotes cysteine generation, and cysteine is a precursor of H_2_S. Depletion of CSE in neurons causes decreased H_2_S, which leads to oxidative stress and mitochondrial dysfunction. Furthermore, it has been reported that the transsulfuration pathway linked to cysteine and H_2_S metabolism is disrupted in HD [10]. Therefore, administration of cysteine through the use of N-acetylcysteine (NAC) can ameliorate the HD-associated excitotoxicity and depressive behaviors by specifically increasing glutamate in a glutamate transporter-dependent manner in the R6/1 transgenic mouse model of HD [126]. Furthermore, the transsulfuration pathway regulates signaling events through sulfhydration which is a posttranslational modification mediated by H_2_S [127,128]. Diminished levels of CSE leads to decreased levels of sulfhydration in striatal cell-line model Q111 of HD, suggesting that H_2_S may be decreased in the brain of HD [129]. In addition, the enzyme CBS affects H_2_S production by specifically interacting with Huntingtin protein [130]. Therefore, it is speculated that H_2_S may be involved in regulating HD, and treatment with H_2_S donors may benefit HD [108]. Recently, it was found that NaHS protect rats against 3-nitropropionic acid (3NP)-induced HD-like pathology through activation of anti-oxidative responses, and anti-inflammatory, as well as anti-apoptotic signals [131]. Moreover, it has been reported that 40 mg/kg H_2_S is clinically beneficial for people with HD disease [132].

### 4.4. Diabetes-Associated Cognitive Impairment and Other Neurodegenerative Diseases

Numerous studies have pointed out that patients with diabetes are more likely to develop cognitive impairment than healthy individuals [133], resulting in a serious social burden [134]. However, as the exact mechanism of diabetes-associated cognitive decline (DACD) is not completely clear, there is no effective treatment for the disease. Recent studies have indicated that ER stress in the hippocampus and synaptic dysfunction contribute to diabetes-related cognitive disorders [135,136,137,138,139]. More importantly, it has been reported that endogenous H_2_S production is decreased during the pathogenesis of diabetes [140,141], suggesting the possible physiological involvement of H_2_S in regulating diabetes-induced cognitive disorders. Accordingly, it was found that administration of NaHS promoted hippocampal endogenous H_2_S production in streptozotocin (STZ)-induced diabetic rat models, and significantly ameliorated the diabetic-associated cognitive disorders by inhibiting ER stress, including preventing expressions of glucose-regulated protein 78 (GRP78), cleaved caspase-12, and C/EBP homologous protein (CHOP) [142]. Meanwhile, He et al. found that NaHS could improve the cognitive dysfunction of STZ-diabetic rats by promoting the expression of SIRT1, a major regulator in synaptic plasticity [143]. In the db/db mice model, it has also been reported that exogenous injection of NaHS could decrease ER stress [144]. Considering that the upregulation of *SIRT1* inhibits the ER stress pathway [145], it is possible that H_2_S could improve diabetes-associated cognitive deficits by inhibiting hippocampal ER stress through upregulating SIRT1. However, the mechanisms through which SIRT1 regulates ER still remain unclear. Because the activity of SIRT1 is associated with mitochondrial biogenesis and mitophagy, the ER and mitochondria are tightly contacted at specific subdomains via tethering mediated by mitochondria-associated ER membrane (MAM) proteins, which are critical for determining cell fate [146]. It can be hypothesized that H_2_S may modulate diabetes-related learning and memory decline by regulating ER-mitochondria signaling. Moreover, it was reported that exogenous H_2_S had anti-apoptotic and anti-inflammatory effects in ameliorating the diabetes-associated cognitive deficit by downregulating mitochondria-mediated pro-apoptotic genes cleaved Caspase-3, cleaved Caspase-9, Bax and cytochrome C, and expressions of IL-23/IL-17 [147]. Therefore, mitochondria-targeted H_2_S donors with the regulation of mitophagy could be used as a new strategy to improve diabetes-associated cognitive impairment.

As is well known, H_2_S exerts its antioxidant functions by ameliorating oxidative stress and preserving mitochondrial function in neurodegenerative diseases [75,108]. However, low physiological concentrations of H_2_S are generally cytoprotective, whereas higher doses are detrimental. For example, amyotrophic lateral sclerosis (ALS) is defined as the selective degeneration of upper and lower motor neurons caused by mutations of Cu/Zn superoxide dismutase 1 (SOD1) genes and dysregulation of mitochondrial complexes II and IV [10,75,148]. Studies performed in spinal cord cell culture lines, familial ALS (FALS) mouse model, and sporadic ALS patients, respectively, showed that H_2_S concentrations were increased in astrocytes and microglia and contributed to motor neuron death by inducing intracellular Ca^2+^ accumulation in motor neurons, and activating Nrf-2-mediated oxidative stress response and peroxiredoxins [149,150]. Interestingly, it was reported that treatment with amino-oxyacetic acid (AOA, a systemic dual inhibitor of CBS and CSE) only expanded the lifespan of female fALS mice [151]. Based on these findings, we can conclude that the therapeutic or deleterious effect of H_2_S on neurodegenerative diseases mainly depends on the type of H_2_S donors and modulators, and the timing of therapy [152,153,154].

**Table 1 antioxidants-12-00652-t001:** Summary of the beneficial effect of H_2_S in neurodegenerative diseases.

Diseases	Biological Model	Targets	References
AD	PC12 cells	Mitochondrial membrane potential, Redox steady state, Apoptosis pathway	[90,91,98]
	APP/PS1 neurons	Energy production, Mitochondrial DNA, Redox steady state, APP pathway	[92,102]
	HepG2/HEK293 cells	Mitochondrial bioenergetics	[93]
	3xTg-AD mice	Redox steady state, GSK3β	[94,95,105]
	APP/PS1 transgenic mice	Nrf2, HO-1, GST	[96]
	Homocysteine-induced AD mice	NMDA receptor	[97]
	Microglia cells	p38 and JNK-MAPK dependent pathway, STAT3, Cat S	[99,101]
	Hippocampus cells	NF-κB pathway	[100]
PD	Rotenone-induced PD mice	Mitochondrial function, JNK-MAPK pathway, anti-inflammatory factors	[82,110]
	SH-SY5Y cells	Mitochondrial transmembrane potential, Cell apoptotic, Redox steady state	[109]
	MPP^+^-induced PD mouse	SIRT1 expression and sulfhydration, PGC-1α deacetylation	[118,120]
	6-OHDA-induced PD rat	Leptin signaling, Redox steady state	[110]
	PD patients	Parkin sulfhydration	[121]
HD	3-nitropropionic acid (3NP)-induced HD rat	Antioxidative responses, anti-inflammatory and anti-apoptotic signaling	[131]
DACD	Streptozotocin (STZ)-induced diabetic rat	GRP78, cleaved caspase-12 and CHOP expression, SIRT1 expression	[142,143]
	db/db mice	SIRT1 expression, ER stress pathway, anti-apoptotic and anti-inflammatory pathway	[145,147]

## 5. Conclusions and Perspective

This review summarized the general toxic substances that contribute to neurotoxicity, and its possible mechanisms resulting in neurodegenerative diseases, including oxidative stress and mitochondrial dysfunction. Furthermore, we described the physiological roles of novel gas transmitter H_2_S in protecting neurons by regulating antioxidative, anti-inflammatory and anti-apoptotic responses (Table 1, Figure 4). Although the mechanism of H_2_S-mediated protein persulfidation has been reported in modulating some neurodegenerative diseases, many unanswered questions remain, from those related to its biological functions to specific target proteins of sulfhydration. Furthermore, most previous studies mainly focused on the beneficial effect of exogenous H_2_S application on neurodegenerative diseases, while overlooking the exploration of exact technology for measuring and modulating its intracellular production and metabolism. In addition, we concluded that the basic endogenous level of H_2_S varies in different tissues, and different neurodegenerative diseases have different requirements for H_2_S and polysulfides.

The impairment of mitochondrial function mainly includes disruptions of mitochondrial enzyme activities, reduction in circulating mitochondrial DNA content, loss of circular structure and dysregulation of biogenesis. Further investigation of the effect of H_2_S on mitochondrial activity in neurodegenerative diseases is expected to advance our understanding of future drug targets. Future research will increase the identification of additional targets of S-sulfhydration, and perhaps also reveal the extended conservation of this process through interacting with other signal pathways in different diseases. In addition, iron dyshomeostasis is implicated in various neurodegenerative diseases. Endogenous H_2_S production is regulated by iron via non-enzymatic reactions, implying the inter-relationship between H_2_S and iron in neuron diseases. However, the exact molecular mechanisms for the interplay between H_2_S and iron, and their therapeutic applications in neuron diseases need to be further explored. Thereby, research on the supplementation of H_2_S-releasing drugs or modulation of its metabolism would be benefit future therapeutic strategies of neurodegenerative diseases in clinic.

Oxidative damage and mitochondria dysfunction are regarded as the primary cause of neurodegenerative diseases, including AD, PD, HD, DACD, and ALS. H_2_S exerts its neuroprotective effects through upregulating (marked in green) or downregulating (marked in red) the antioxidant, antiapoptotic, and anti-inflammation genes, or influencing activities of specific proteins through sulfhydration. Strictly controlling the endogenous H_2_S production through the administration of mitochondrial-targeted H_2_S donors and modulation of enzymatic or non-enzymatic mediated H_2_S metabolism would be beneficial for antioxidative stress and preserving mitochondria activity and biogenesis, thus subsequently improving neurological diseases. This model was created using BioRender (https://biorender.com/, accessed on 20 November 2022).

## Figures and Tables

**Figure 1 antioxidants-12-00652-f001:**
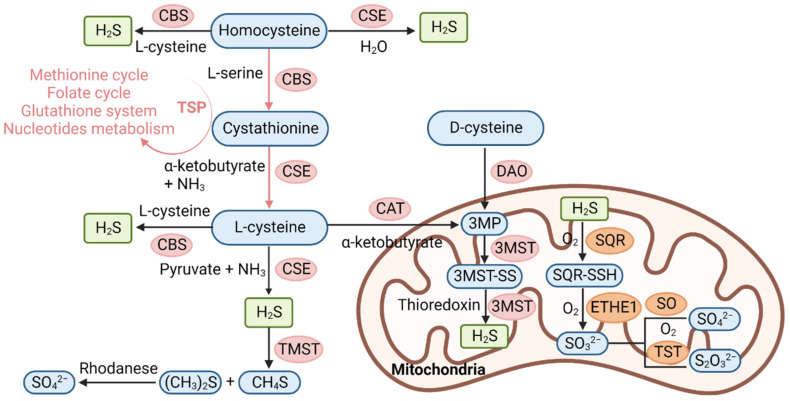
Cellular biosynthesis and oxidation of H_2_S.

**Figure 2 antioxidants-12-00652-f002:**
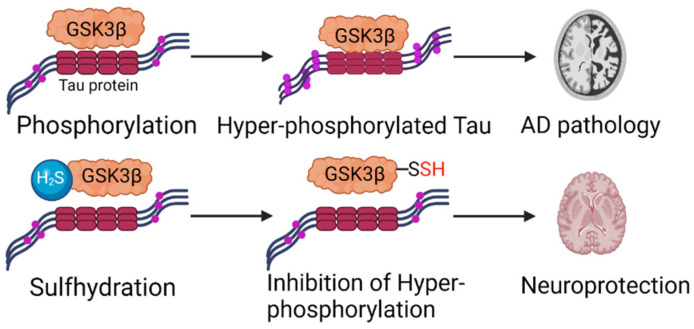
Proposed model of neuroprotection of AD afforded by H_2_S.

**Figure 3 antioxidants-12-00652-f003:**
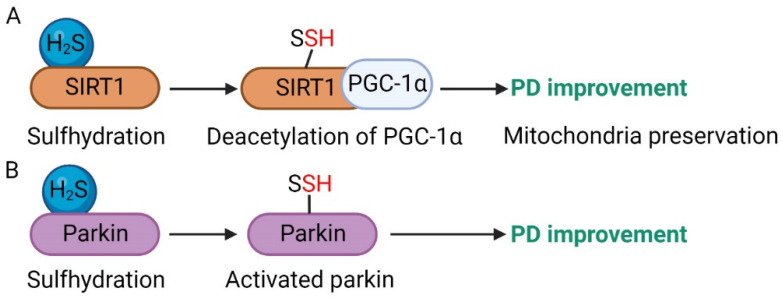
Proposed neuroprotective mechanisms of H_2_S in regulating PD. (**A**) H_2_S sulfhydrates the SIRT1 and promotes the activation of SIRT1, which subsequently activates the deacetylation of PGC-1α, resulting in mitochondria preservation and PD improvement. (**B**) H_2_S promotes the activation of parkin through sulfhydration, which finally leads to PD improvement. The red fonts represent sulfhydryl group. This model was created using BioRender (https://biorender.com/, accessed on 19 November 2022).

**Figure 4 antioxidants-12-00652-f004:**
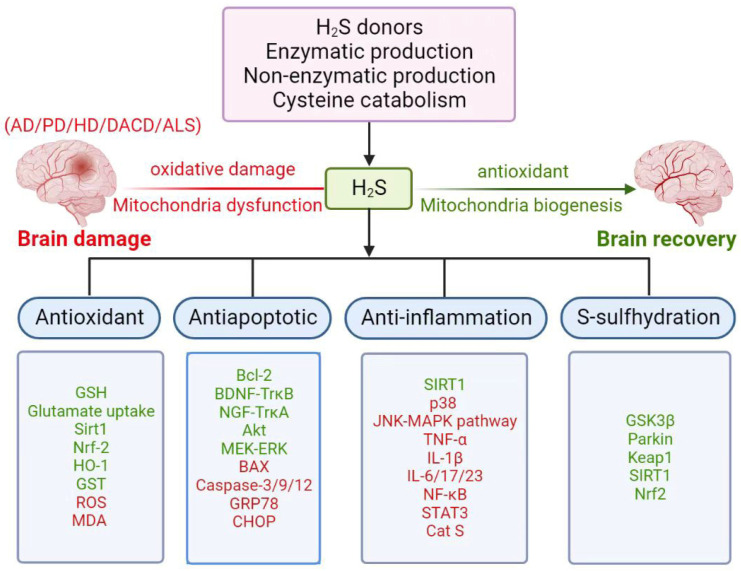
Overview of H_2_S in regulating neurodegenerative diseases.

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
