# Peer review of "Advances of H2S in Regulating Neurodegenerative Diseases by Preserving Mitochondria Function"

_antioxidants, 2023, doi:10.3390/antiox12030652_

Round 1

Reviewer 1 Report

 In this manuscript the authors focus on review the role of H2S in the develop neurodegenerative diseases through its function as an anti-oxidative, anti-inflammatory, and anti-apoptotic factor. This topic has been previously review (for instance (Goodwin et al. (2019) , Determination of sulfide in brain tissue by gas dialysis/ion chromatography: postmortem studies and two case reports J Anal Toxicol1989), however in this manuscript the authors review the recent literature, including  studies in which are analysed the role of H2S as a possible  therapeutic molecule for neurological diseases. I therefore consider that this work contains interesting and valuable information. However, I find some problems that I think the authors should address.

-The text contains some paragraphs and sentences that are difficult to understand and confusing. For example, the sentence (line 28') ' In brain nervous system, enormous consumption of oxygen and ATP is mainly dependent on mitochondria “could be interpreted as meaning that the major ATP and Oxygen consumers in the nervous system are mitochondria. However, I think the authors mean that the mitochondria are the main producer of ATP.

There are other phrases such as in line 25 “neurodegenerative diseases in both humans and animals are usually complicated” in which it is not clear what the authors mean. There are more examples throughout the text.

I have found also several paragraphs that are confusing for instance, between lines (276 – 282), (348-351) and (395-398). The authors should carefully revise the text and clarify these paragraphs and sentences.

-The authors do not include in figure 1 the mechanism of oxidation of H2S through methylation via thiol S-methyltransferase (TSMT)

-In the section 4.1. Alzheimer’s disease (AD). In my opinion it would be interesting if the authors include papers in which it has been suggested that soluble AB can deregulate H2S signaling via inhibition of neural cysteine transporter EAAT3/ EAAC1 resulting in oxidative stress and neurodegeneration (Hodgson et al (2013) J Alzheimer Dis 36; Auyama et l. (2006) Nat Neurosci.9)

-The initial part of the 4.2. section is quite repetitive with parts of previous sections. In general, this section is confusing and in many cases is a succession of unconnected results. I think the authors should revise it and try to make it more understandable and readable.

Reviewer 2 Report

In this manuscript Lina Zhou and Qiang Wang review the role of H2S in neuroprotection through anti-oxidative, anti-inflammatory, anti-apoptotic and S-sulfhydration, and highlights the importance of H2S as a therapeutic molecule for neurological diseases.

Comments

The manuscript. It needs extensive English editing. For example:

In the Abstract: Nervous system is extremely vulnerable to oxidative stress because of its high demand of oxygen and biogenesis.

Biogenesis of what?

Given the relevance of iron accumulation in many neurodegenerative diseases it would be of interest to include section about the interplay between H2S and iron.

It would be informative yo include a table with information about diseases, biological model and references reporting the beneficial effect of H2S donors.

Information about current clinical trials evaluating the effects of H2S donors in neurodegenerative diseases would be of interest.

Reviewer 3 Report

In this paper Zhou et al discuss the beneficial effects of H2S on neurodegenerative diseases. The paper is written in broken English and some of the sentences are not clear and left to the reader interpretation. They should fix this issue because it is hard to understand and follow what the authors are trying to say. Also, there are other things that need to be addressed:

·   Figure 2 and 3: These figures only represent one of the targets of H2S in AD and PD. It is better if they add a table per each disease describing the major targets of H2S based on the studies they discussed in each section.   

 ·    Add a table also to describe the effect of H2S in Huntington.

 · At the end of section 4.4 they briefly discuss the role of H2S in ALS another major neurodegenerative disease in which mitochondrial dysfunction plays a pivotal role. Therefore, for completion they should add another section describing all the known studies on H2S and ALS in vivo and in vitro even if they have a different outcome compared to the other disease discussed.

Round 2

Reviewer 1 Report

The authors have addressed all my comments

Reviewer 2 Report

The authors have addressed all my concerns

Reviewer 3 Report

The authors have addressed in fully all my concerns.